# A Study on the Literacy Rate of Buddhist Sangha in the Tang Dynasty

**Shaowei Wu**

School of History, Shandong University, Jinan 250100, China; wushaowei5719@163.com

**Abstract:** The Buddhist sangha played a crucial role in ancient China, exerting significant influence on its society through religious identity and cultural knowledge. However, not all members of the monastic community were literate. The Tang Dynasty introduced an examination system that assessed monks' proficiency in reciting Buddhist scriptures, determining their eligibility for ordination. Simultaneously, efforts to remove unqualified monks and nuns provided an opportunity to estimate the literacy rate within the monastic community. A statistical analysis of the literacy rate offers a novel perspective for understanding the evolution of Buddhism, the intricate relationship between religion and politics, and the role of the monastic community in local society during the Tang Dynasty.

**Keywords:** Tang Dynasty; Buddhist sangha; literacy rate; scripture recitation examination

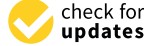



The Buddhist sangha constituted a significant cultural entity in ancient Chinese society, exerting profound influence. The elite segment within the monastic community, particularly those high-ranking monks in closer proximity to imperial authority, played a pivotal role in shaping national Buddhist policies and influencing the dynamics of the state–religion relationship. Previous studies predominantly focused on these monks. Nevertheless, it is imperative to underscore that the comparatively overlooked lower and middle-ranking monks within the monastic community constitute a more sizable contingent, engage more extensively with the general populace, and wield a more pervasive and quotidien influence. It was precisely due to their endeavors that, during the medieval period, Buddhist monasteries evolved into pivotal cultural centers within regional societies. The manifestation of such an influence by the sangha stems not only from their religious authority but also from their cultural knowledge (Xie 2009). Erik Zürcher once described the ancient Buddhist sangha as "the Secondary Elite" (Zürcher 1989). Consequently, the investigation of their cultural attributes, particularly the literacy rate within the monastic community, becomes of paramount importance[1].

## 1. Not All Monastics Possess Literacy

Regarding the issue of literacy rates within the Buddhist sangha, there has been a lack of rigorous scholarly investigation in previous studies. When discussing literacy rates in modern times, some scholars even hastily classify all religious groups, including Buddhism and Taoism, directly into the literate category (Mote 1972). Perhaps this impression arises because many activities undertaken by the sangha are inherently connected to written texts. Internally within Buddhism, a significant task for ordained individuals is the recitation of scriptures, which undoubtedly imposes literacy requirements on members of the sangha. According to Vinaya, such requirements are articulated from the moment of ordination. Novices (śrāmaṇeras) are mandated to recite scriptures daily[2], with literacy being a presumed prerequisite. The internal mechanisms of Buddhism ensure a cultural demand within the sangha, compelling its members to be more motivated to be literate compared to lay practitioners. During the medieval period in China, the state repeatedly issued decrees mandating that ordained monks within the sangha must possess a certain

level of proficiency in reading scriptures. However, whether articulated in Vinaya regulations or state decrees, they undeniably present an idealized state, and the practical situation may significantly deviate from these standards. Furthermore, from another perspective, the recurrent issuance of state decrees implies the existence of individuals within the sangha who were unable to read scriptures.

*The Sūtra of the Dharma's Complete End* (*Fa miejin jing* 法滅盡經), an apocryphal sutra dated to the end of the 5th century, revealed that many monks at that time "did not chant scriptures" 經不誦習 and "did not know characters and sentences" 不識字句[3]. In the first year of the Yuanxing 元興 era (402), during the reign of Emperor An of the Jin Dynasty 晉安帝 (382–419, reg. 397–419), the Grand Commandant Huan Xuan 桓玄 (369–404) perceived a decline in Buddhism. He believed that there were too many inadequately qualified monks and nuns, necessitating a purge. Consequently, he promulgated an ordinance for the expulsion of monks and nuns (料簡沙門書), outlining specific criteria for elimination. According to the decree:

> Those monks who possess the ability to articulate the doctrinal principles of Buddhist scriptures, those who rigorously adhere to the precepts of Buddhism without any transgressions, and those who dwell in mountainous and forested areas refraining from lay engagements are considered conducive to the dissemination of Buddhism. They contribute to the governance of the lay society and serve as exemplary models for individuals. Apart from this subset of monks, all other monastics are mandated to return to lay life and assimilate into the ordinary populace. 諸沙門有能申述經誥，暢說義理者；或禁行修整，奉戒無虧，恒為阿練若者；或山居養志，不營流俗者，皆足以宣寄大化，亦所以示物以道，弘訓作範，幸兼內外。其有違於此者，皆悉罷遣。[4]

These criteria, particularly the qualification of having "the ability to articulate the doctrinal principles of Buddhist scriptures", evidently target those monks who are unable to recite and understand the Buddhist scriptures. This indicates that Huan Xuan was acutely aware of the presence of illiterate monks within the monastic community during that period. Nevertheless, they took into consideration other virtues of the monastic community, recognizing that monks' ethical conduct is a crucial factor enabling their ordination and entry into monastic life. Hence, they established multiple criteria to account for these various virtues.

At the time that the Emperor Wu of Northern Zhou 北周武帝 (543–578, reg. 560–578) tried to exterminate Buddhism, he once required those who were illiterate to return to laity. He posited that "inadequately qualified monks might refrain from voluntary lay reversion due to feelings of shame" 寡德沙門恥還於素俗. Consequently, he established explicit screening methods and criteria for monks. Specifically, he mandated local officials to conduct examinations, and based on the assessment outcomes, only those monks capable of reciting Buddhist scriptures and having undergone multiple checks were permitted to remain within the community. Namely, the defrocking of monks and nuns was accomplished through the recognition and recitation of classical scriptures, compelling those unable to read to return to lay life. This shares similarities with Huan Xuan's approach; however, the difference lies in that the Emperor Wu established only this criterion, disregarding the other virtues of monks. After the promulgation of the decree, it faced vehement opposition from monks such as Tanji 曇積. Tanji emphasized the diverse virtues possessed by different monks and enumerated five reasons to oppose the Emperor Wu's policy. However, concurrently, he admitted that there were many monks who were "obtuse, and lacking a gift for reading. They study hard but have not learned a single character" 受性愚鈍，於讀誦無緣；習學至苦，而不得一字. "They are not smart and they cannot read more than one phrase" 無聰力，日誦不過一言. Some monks practicing asceticism were also incapable of "reciting." [5] In accordance with TanJi's petition, his opposition was not directed at the defrocking per se but rather at the criteria for defrocking. He contended that it is imperative to differentiate the diverse virtues of individual monks. This proposition aligns relatively closely with the views of Huan Xuan.

In the later period of the Southern Chen dynasty during the reign of the Last Emperor 陳後主 (553–604, reg. 582–589), there arose a situation discovered during the reorganization of the sangha, wherein "there were as many as tens of thousands of monks and nuns without proper monastic credentials" 無貫者萬計 In response to this, many court officials proposed the defrocking of the monastic population and advocated for a method of "examining the recitation of scriptures", stipulating that "monks and nuns who fail the scripture recitation test should return to lay life" 策經落第者，並合休道. Although this proposal was not implemented due to opposition from the wise master Zhiyi 智顗 (538–597)[6], it nevertheless indicates that contemporary authorities were aware of a significant number of illiterate monks in the Southern Chen dynasty.

During the early Tang Dynasty, there persisted individuals who were illiterate. Notably, Huineng 惠能 (638–713), the renowned founder of the Southern School of Chan Buddhism, even said that he had "not recognized characters since birth" 一生以來，不識文字. While the factual accuracy of this claim may be subject to debate, it underscores that, in his perspective, illiteracy was not perceived as a hindrance to monastic life in a monastery. Illiteracy was not an isolated phenomenon but endured for an extended period. By the Song Dynasty, there still existed a considerable illiterate demographic. Pro. Minsheng Cheng (2019) even suggested that it might have been as high as one-third, although he did not provide a specific basis for this estimation. Nevertheless, the existence of a significant number of illiterate monks and Daoists during the Song Dynasty is an historical reality.

In general, the Buddhist sangha in ancient China was not entirely literate. However, before the Tang Dynasty, to be more precise, prior to the reign of Emperor Gaozong of Tang 唐高宗 (628–683, reg. 649–683), it is challenging to precisely gauge the literacy rate of the monastic community. This is primarily due to the diverse avenues through which individuals could qualify for monastic ordination during this period, and the state's attitude toward illiterate monks was not uniformly negative. Although Emperor Wu of Northern Zhou once attempted to employ literacy as a criterion for scrutinizing and reorganizing monks and nuns, whether this measure was fully implemented lacks explicit documentation in historical records. Since the reign of Emperor Gaozong of Tang, the potential for calculating the literacy rate of the sangha emerged due to a dual phenomenon during this period. Firstly, there was a gradual normalization of state-led population censuses for the sangha[7], resulting in the preservation of substantial data on the number of monks and nuns during the Tang Dynasty in historical records. Secondly, from the reign of Gaozong onward, literacy among monks became an official means of controlling the sangha. The institutional framework mandated that all individuals seeking ordination must pass a scripture recitation examination. In theory, the implementation of this policy could ensure literacy for all those receiving monastic ordination. However, historical evidence indicates that clarifying measures targeting monks and nuns were undertaken in nearly every dynasty, suggesting the persistence of illiterate monks. However, for our current research, what is crucially significant is that certain activities related to the defrocking of monks and nuns have preserved defrocking data. This facilitates the potential for statistical analysis and an estimation of the number of illiterate monks within the sangha.

## 2. The Scripture Recitation Examination System Implemented within the Buddhist Sangha during the Tang Dynasty

Since the Southern and Northern Dynasties, the government has consistently sought to strengthen control over the size of the Buddhist sangha, and one of the measures employed is the establishment of the ordination system[8]. However, during the period of the Eastern Jin and Southern and Northern Dynasties, the criteria for monastic ordination were relatively broad, and individuals could qualify for monastic ordination through various means. At that time, the government placed particular emphasis on the moral character of applicants. By the Tang Dynasty, the criteria for ordination became increasingly specific, and the methods became more diverse, including scripture recitation examination (*shijing* 試經), special policies decreed by the emperor, and purchasing monastic ordination with

money. Among these, *shijing* emerged as the most important and prevalent method of ordination in the early Tang period. *Shijing* involves the recital of Buddhist scriptures, and only those who recite a certain quantity of scriptures qualify for monastic ordination[9]. This shift from emphasizing the moral character of applicants to emphasizing their cultural proficiency represents a significant transformation in the ordination system during the Tang Dynasty.

The formalization of the scripture recitation examination gradually occurred no later than the Xianqing era 顯慶 (656–661) under Emperor Gaozong[10]. The earliest recorded *shijing* took place in the third year of Xianqing (658). In that year, "the emperor decreed, selecting 150 boys for monastic ordination through an examination that tested their memorization of Buddhist scriptures" 令詮試業行童子一百五十人擬度[11]. Subsequently, after Gaozong, almost every dynasty issued explicit decrees regarding the examination of scripture recitation for monastic ordination. For instance, in the second year of the Shenlong 神龍 era (705), Emperor Zhongzong 唐中宗 (656–710, reg. 684, 705–710) issued a decree stating, "Nationwide examinations on Buddhist doctrinal knowledge will be conducted for those boys and practitioners preparing to apply for monastic life. Individuals demonstrating a profound understanding of Buddhist doctrines will be selected and granted monastic ordination" 天下試童行經義，挑通無滯者，度之為僧[12]. Despite the weakening of state control over monks and nuns after the An Lushan Rebellion 安祿山叛亂, decrees continued to be issued regarding the scripture recitation. In the second year of the Zhide 至德 era (757), Emperor Suzong 肅宗 (711–762, reg. 756–762) ordered, "lay individuals who can read five hundred sheets of Buddhist scriptures will be eligible for monastic ordination under the pretext of understanding Buddhist scriptures" 白衣誦經五百紙，賜明經出身為僧[13]. In the eighth year of the Dali 大曆 era (773), Emperor Daizong 唐代宗 (726–779, reg. 762–779) stipulated, "Nationwide examinations in the three categories of scriptures, monastic rules, and treatises will be conducted for boys and practitioners intending to apply for monastic life. [Only those who pass the examinations] will be eligible for monastic ordination" 敕天下童行策試經律論三科，給牒放度[14].

The standards for the scripture recitation examination varied across different periods. For example, during the reign of Emperor Xuanzong 玄宗 (685–762, reg. 712–756), the requirement was set at "two hundred sheets" 二百紙[15], which was later increased to seven hundred sheets during the reign of Emperor Suzong[16]. In the Wenzong era, the requirement specified that "for monks and nuns during the examination, the ability to read five hundred sheets of scriptures is considered qualified" 試經僧尼，必須讀得五百紙[17]. The Tang Dynasty had its lowest requirement during the reign of Emperor Jingzong 敬宗 (809–827, reg. 824–827), who specified that "monks who can recite one hundred and fifty sheets of Buddhist scriptures and nuns who can recite one hundred sheets are eligible for monastic ordination" 僧能暗誦一百五十紙、尼一百紙，即令與度[18]. If we consider the official standard of 28 lines per sheets and 17 characters per line during the Sui and Tang periods, monks and nuns needed to memorize approximately 71,400 characters and 47,600 characters, respectively. In terms of the monk's standard, this roughly equates to a portion of the *Lotus Sutra* 法華經.

Those who passed the examination were eligible for monastic ordination and were recorded in the register of monks and nuns 僧尼籍. In the register, it was necessary to specify the scriptures they had studied. For instance, a memorial submitted to Emperor Wenzong 唐文宗 (809–840, reg. 826–840) by the Ministry of Rites in the fourth year of the Dahe era (830) described the format of the monk and nun register at that time.

Henceforth, for those monks and nuns who passed the examination, the local officials should separately record their monastic names, lay surnames, places of origin, the masters listed in their lay household registers, the academic achievements of these monks and nuns, and the number of sangha assigned to each temple. The compiled register of monks and nuns should be submitted to the Ministry of Rites for subsequent verification. 起今已後，諸州府僧、尼已得度者，勒本州府具法名、俗姓、鄉貫、戶頭、所習經業及配住寺人數，開項分析，籍帳送本司，以明真偽.[19]

In the second year of the Xiande 顯德 era (955) of the Later Zhou Dynasty, the court stipulated:

> One month before the emperor's birthday, [local officials] must record the names, places of origin, assigned temples, ages, and academic achievements of those eligible for monastic ordination in the register of monks and nuns. The register is then to be submitted to the Ministry of Rites. The Ministry of Rites will issue monastic ordination certificates based on the register, after which these individuals are allowed to undergo monastic ordination. 一應合剃頭受戒人等，仰逐處於天清節一月前，具'姓名、鄉貫、寺院、年幾，及所習經業'申奏，候勑下，委祠部給付憑由，方得剃頭受戒.[20]

In addition to the *shijing* during monastic ordination, in the early Tang Dynasty, the state implemented a relatively rigorous daily scripture recitation examination system. Specifically, when compiling the register of monks and nuns, a scripture recitation examination was conducted. In the twelfth year of the Kaiyuan era (724), during the reign of Emperor Xuanzong, regulations were established for the *shijing*, requiring monks to "read two hundred sheets of Buddhist scriptures, with a minimum of seventy-three sheets read annually, and an examination conducted every three years" 誦二百紙經，每年限誦七十三紙，三年一試[21]. It is noteworthy that the mention of "an examination conducted every three years" aligns with the contemporaneous regulation of compiling the register of monks and nuns every three years. This indicates a close relationship between the *shijing* and the compilation of the register.

Scholars have reconstructed the edict issued in the twenty-fifth year of Kaiyuan, found in the *Tiansheng ling* 天聖令, which pertains to the "compilation of the register of monks and nuns". According to this regulation:

> Eevery three years, a comprehensive census of all Taoists, female Taoists, monks, and nuns was to be conducted at the state and county levels, resulting in the compilation of a register. The register was required to include information such as the time of their monastic qualification, duration of monastic life, and academic achievements. The seals of the state and county were to be affixed to the designated positions on the register. 諸道士、女冠、僧、尼，州縣三年一造籍，具言出家年月、夏臘、學業，隨處印署.[22]

The term "academic achievements" mentioned in this regulation likely corresponds to the *shijing* referred to in the law of the twelfth year of Kaiyuan. The practice of conducting scripture recitation examinations during the compilation of the register of monks and nuns did not originate with Emperor Xuanzong but had already become an established system by the time of Emperor Gaozong. An example from the unearthed *Monk Register of the Sien Temple in Gaochang County, Xizhou, in the Second Year of Longshuo (662) of the Tang Dynasty* 唐龍朔二年（662）西州高昌縣思恩寺僧人戶口簿 in Turpan, Xinjiang, dated 2004, indicates that each monk's academic achievements were recorded. For instance, Monk Zhang had read five volumes of the *Lotus Sutra*, one volume of the *Yaoshi jing* 藥師經, and one volume of the *Foming jing* 佛名經, while Monk Chongdao 崇道 had read five volumes of the *Lotus Sutra*[23]. These academic achievements evidently did not encompass the entire content studied during the three-year period but likely reflected the recitation undertaken during the examination before the compilation of the register of monks and nuns.

In the Tang Dynasty, the requirements for monks to obtain monastic ordination were stringent, coupled with regular examinations. These institutional mechanisms ensured a relatively high cultural standard within the Buddhist sangha. However, there were instances of inadequately qualified monks from time to time. In the early Tang period, concerted efforts were made to rectify this situation. Although historical records do not explicitly document specific elimination criteria, it is likely that the *shijing* played a crucial role in this endeavor. For instance, during the reign of Emperor Xuanzong, an edict was issued:

> Nationwide, all monks and nuns under the age of 60 were mandated to undergo a scripture recitation examination. Each individual must recite 200 sheets of Bud-

dhist scriptures. Those monks and nuns who failed to meet the specified requirements were obligated to return to lay life. Additionally, alternative forms such as meditation (坐禪) or expounding Buddhist scriptures were explicitly prohibited as substitutes for the scripture recitation examination. 有司試天下僧尼年六十以下者，限誦二百紙經，……落者還俗，不得以坐禪、對策義試.[24]

Emperor Xuanzong stipulated that disqualified monks and nuns must return to lay life and emphasized the exclusive use of the scripture recitation examination as the assessment method, precluding alternatives like meditation or scripture exposition. The primary aim was evidently to evaluate the literacy and recitation abilities of monks and nuns. Emperor Xuanzong further mandated the establishment of roles such as *Qingdi shangzuo* 清滌上座 (sthavira) in temples nationwide, tasked with improving the academic standards of monks and nuns within their respective temples[25]. In another instance, in the ninth year of Dahe 大和 (835), Li Xun 李訓 presented a memorial to Emperor Wenzong, expressing concern over the large number of inadequately qualified monks and nuns consuming significant state and social resources. Subsequently, with Emperor Wenzong's support, a nationwide scripture recitation examination for monks and nuns was instituted, and those who failed were forcefully returned to lay life[26].

The monks and nuns eliminated through the scripture recitation examination were predominantly illiterate individuals. In other words, the number of disqualified monks and nuns likely closely mirrored the size of the illiterate population. This provides a potential basis for estimating the literacy rate within the monastic community[27].

### 3. Estimation of Literacy Rates within the Buddhist Sangha in Different Periods of the Tang Dynasty

The statistical estimation of the literacy rate within the Buddhist sangha involves, on one hand, the relatively accurate total figures of the sangha population and, on the other hand, the relatively accurate numbers of literate or illiterate monks. Historical materials provide several records of the size of the Buddhist sangha in different periods of the Tang Dynasty, offering data for the total number of monks. Additionally, information from various purges of monks and nuns during the Tang Dynasty provides insights into the elimination of certain individuals, contributing to the potential estimation of illiterate monks. Therefore, by combining these two sets of data, we can make a rough estimate of the literacy rate within the Buddhist sangha during different periods of the Tang Dynasty.

### 3.1. Literacy Rate of the Buddhist Sangha from the Reign of Emperor Gaozong to Emperor Xuanzong

During the reigns from Emperor Gaozong to Emperor Xuanzong, the state adopted the method of scripture recitation examinations to determine which monks could obtain monastic ordination. Additionally, regular scripture recitation examinations were conducted during the compilation of monk registers, significantly reinforcing governmental control over the Buddhist sangha and ensuring its cultural standards. During this period, policies were rigorously enforced, ensuring the quality of the sangha. The "*Record of Monks of Chongfu Temple Chanting Scriptures in May of the First Year of Zhengsheng* 證聖 (695)" 武周證聖元年(695)五月西州高昌縣崇福寺轉經歷 (73TAM193: 37a, 27a, 30a, 29a, 1a, 27b, 37b, 30b) revealed that monks such as Xuanpan 玄判, Xuanshi 玄式, and Xuanfan 玄範 had recently obtained monastic ordination but were already participating in scripture recitation activities[28]. This indicates that they possessed a certain level of literacy and scripture recitation abilities at the time of their ordination. Therefore, although there were monks like Huineng who were illiterate, they were likely in the minority, and the literacy rate within the sangha during this period appears to have been quite high.

However, the system of strictly using scripture recitation examinations for monastic ordination was temporarily disrupted during the reigns of Empress Wu Zetian to Emperor Ruizong, giving rise to a phenomenon:

Individuals attempting to evade taxes, conscription, and punishment sought refuge in Buddhist temples, effectively becoming unauthorized monks unregistered in the official monk registry. The population of such unauthorized monks reached tens of thousands. [29] 逃丁避罪，並集法門，無名之僧，凡有幾萬.

The total population of the sangha during that time is not explicitly recorded in historical records. The closest available data are from the year 736, where the number of monks and nuns was 126,000[30]. During the reign of Emperor Xuanzong, control over the Sangha's numbers was stringent. In the 19th year of the Kaiyuan era (731), an edict from Emperor Xuanzong mentioned that the emperor had prohibited the ordination of monks and nuns, a prohibition that lasted nearly 20 years[31]. 不度人來，向二十載. This implies that no one was permitted to undergo monastic ordination from Emperor Xuanzong's accession in 712 until 731. Therefore, the data from the year 736 should be close to the data from the early Kaiyuan era. In addition, in the second year of the Kaiyuan era (714), in response to the large number of unqualified monks in the sangha, Yao Chong 姚崇 (650–721) submitted a memorial to Emperor Xuanzong calling for the rectification of the sangha. Different historical sources provide varying accounts of the number of unqualified monks expelled during this rectification, such as over 12,000 according to *The Biography of Yao Chong in Jiu Tangshu*[32], over 20,000 according to *The Annals of Emperor Xuanzong in Jiu Tangshu*[33], and over 30,000 according to *Tang Huiyao*[34]. According to *the epitaph of Helan Wuwen's tomb* 賀蘭務溫墓誌, during the reign of Emperor Ruizong, Helan Wuwen (656–721) also conducted a census of the number of unqualified monks. According to his investigation, there were over 20,000 unqualified monks at that time, requiring compulsory return to lay life[35]. These data align more closely with the account in the *The Annals of Emperor Xuanzong in Jiu Tangshu* and may be closer to the actual situation. Thus, at the end of the reign of Emperor Ruizong, the number of monks in the Sangha was likely around 146,000.

During the reigns from Emperor Taizong 太宗 to Emperor Gaozong, a span of over 60 years, the combined number of monks and nuns barely exceeded 60,000[36]. Surprisingly, within the subsequent thirty years from Empress Wu to Emperor Ruizong, the size of the sangha more than doubled. This period witnessed a notable influx of individuals, many of whom, despite not having undergone scripture recitation examinations, obtained the qualification for ordination and joined the monastic community. The cultural proficiency of these inadequately qualified monks and nuns was decidedly inferior to that of their predecessors in the monastic community.

In the period of Empress Wu, Su Gui 蘇瓌 (639–710) once submitted a memorial mentioning that "the number of inadequately qualified monks and nuns might have accounted for approximately half of the total population" 天下僧尼濫偽相半[37]. This implies that the literacy rate within the sangha during that time could have been around 50%. Although Su Gui's assessment is subjective and cannot provide precise literacy rate data, it does illustrate a significant presence of inadequately qualified monks and nuns. Di Renjie 狄仁傑 (630–704) also reported that there were tens of thousands of monks who had not obtained monastic ordination through legal channels. Based on investigations conducted in the capital, the number of such monks had reached several thousand[38]. Regarding the number of inadequately qualified monks, the more precise statistical data mentioned earlier pertain to the expulsion in 714, totaling over 20,000 individuals. It can be deduced that the proportion of expelled monks and nuns to the total sangha population was roughly 13.7% (12,000/146,000). Therefore, the literacy rate within the sangha was approximately 86.3%. This figure, higher than Su Gui's subjective impression, suggests that despite a considerable number of inadequately qualified monks during the period from Empress Wu to Emperor Ruizong, overall, the imperial control over the sangha remained relatively effective.

With the renewed strengthening of Emperor Xuanzong's control over the Buddhist ecclesiastical community, the literacy rate within the sangha likely experienced a further improvement. The rectification movement initiated in 714 did not conclude within a short period. Based on the *Inscriptions on the Memorial Stone Engraved on the Three Sacred Statues*

三尊真容像支提龕記, dated to the nineteenth year of Kaiyuan (731), the *Qingdi shangzuo* was kept in many temples nationwide, suggesting that Xuanzong's measures for the regulation of monks and nuns persisted for an extended period. From this stone inscription, it is observed that monks frequently emphasized their scripture recitation practices. For example, the inscription highlights the diligent scripture recitation of Monk Yihong 義泓, stating that he diligently studied, reciting the *Vimalakirti Sutra* 維摩詰經 and the *Lotus Sutra* daily. 精勤攝念，策勵持經，《維摩》《法華》，日誦一遍. It also mentions Monk Qianshou 乾壽's profound understanding of Buddhist scriptures, such as the *Lotus Sutra*, *Vijñaptimātratāsiddhi Śāstra* 唯實論, *Abhidharmakosa-sastra* 俱舍論, and *Nyāyamukha* 因明論. 學《法華經》《唯識》《俱舍》《因明》等論，皆理極精微，妙窮法相. While these contents are inherently part of the daily learning routine for monks, their prominent emphasis in inscriptions suggests a close association with the Buddhist policies of the Xuanzong era, particularly the scripture recitation examinations[39].

Emperor Xuanzong's rigorous control over monastic ordination and the frequent scrutiny of individuals entering the sangha systematically closed the loopholes for illiterate individuals seeking refuge in the monastic community. Phenomena similar to Huineng's assertive proclamation during the reigns of Emperor Gaozong and Empress Wu, where he declared, "not recognized characters since birth", likely became less prevalent during this period. Therefore, the literacy rate within the sangha during the Xuanzong period should have shown a noticeable increase compared to the preceding period, possibly approaching a level where almost all monks and nuns were literate.

### 3.2. Literacy Rate of the Buddhist Sangha from the Aftermath of the An Lushan Rebellion

After the outbreak of the An Lushan Rebellion, the state's control over the Buddhist sangha became less effective. In order to raise funds for the military, Emperor Suzong initiated a large-scale encouragement for the populace to obtain monastic ordination qualifications by donating money to the government. Individuals who had "never read any Buddhist scriptures and were completely illiterate" could enter the monastic community by paying additional fees. 未曾讀學、不識文字者[40]. Consequently, the cultural proficiency of the sangha gradually declined. There are few records in historical texts that mention the number of monks who obtained monastic ordination qualifications through monetary donations. For example, in the year 757, the *Jiu Tangshu* recorded, "After the recovery of the two capitals, Chang'an and Luoyang, Emperor Suzong ordered the implementation of the policy of selling monastic ordination certificates in several surrounding provinces of Chang'an. More than ten thousand people obtained monastic qualifications through monetary donations" 及兩京平，又於關輔諸州，納錢度僧道萬餘人[41]. However, these records are quite vague and offer limited assistance in estimating the literacy rate within the sangha.

Fortunately, the Dunhuang documents have preserved information about the sale of monastic ordination certificates during the reign of Emperor Suzong, providing us with a relatively accurate understanding of the situation in the six provinces of Hexi (Liangzhou 涼州, Ganzhou 甘州, Suzhou 肅州, Guazhou 瓜州, Shazhou 沙州 and Yizhou 伊州). Dunhuang Document P. 4072(3), titled "*Report on the Sale of Tonsure Certificates in Hexi by Zhang Jiali in the Second Year of Qianyuan (759)*" 唐乾元二年（759）張嘉禮河西六州納錢度僧告牒, records a total of 327 monks and 169 nuns receiving monastic ordination in the six provinces of Hexi. By the end of the Kaiyuan era, there were 3245 monasteries nationwide with 75,524 monks and 2113 nunneries with 50,576 nuns[42]. With 328 provinces in the country, this averages 10 monasteries per province with an average of 23 monks and 6 nunneries per province with an average of 24 nuns. Therefore, in the six provinces of Hexi, there were approximately 60 monasteries, 36 nunneries, 1380 monks, and 864 nuns. Thus, the number of monks and nuns who obtained monastic ordination through monetary donations in 759 accounted for around 19.2% (327/1707) and 16.4% (169/1033) of the total population in monasteries and nunneries, respectively. This suggests that even if all these individuals were illiterate, the literacy rates at that time might have been approximately 80.8%

for monks and 83.6% for nuns. These data also corroborate the literacy rate of the local Buddhist sangha in Dunhuang during the early period of Tubo rule. Tubo was an empire established by Tibetans from the early 7th century to the mid-9th century, and it ruled Dunhuang from 786 to 848.

The Dunhuang manuscript S.10967, housed in the British Library and titled "*Record of Scripture Chanting among Monks in the Lingtu Monastery and Other Monasteries Around the Year 789*" 789年前後靈圖寺等寺僧眾轉經歷, is a document that lists participants in a chanting event for blessings. The document records the names and assigned tasks of 34 monks from 10 different monasteries. Among these, Lingtu Monastery had the highest number of monks, with 13 participants, while other monasteries had a maximum of 3 participants each. Therefore, this event was primarily led by monks from Lingtu Monastery. The chanting monks were undoubtedly literate, and based on document S.2729(1), "*Report on Population Statistics Book of the Sangha Tribe of Shazhou under the Mi Jingbian in the Third Month of the Year of Dragon (788)*" 辰年(788)三月沙州僧尼部落米淨辯牒上算使勘牌子曆, Lingtu Monastery had 17 monks in March 788. Hence, the literacy rate at Lingtu Monastery during this time was at least 76.5% (13/17). While these data are specific to one monastery, studies suggest that they are similar to other monasteries in the region[43]. The Tubo occupation of Dunhuang began in 786, so by 788, the main members of the Dunhuang sangha were evidently a continuation of the Tang Dynasty sangha. The literacy rate during the An Lushan Rebellion in the Hexi region seems comparable to the early Tubo occupation, indicating a literacy rate of approximately 75% to 80% during the Suzong era. Although relatively high, this rate is significantly lower than that of the Xuanzong era.

After Emperor Suzong, especially following the suppression of the An Lushan Rebellion, subsequent emperors, such as Emperor Daizong 代宗, Emperor Dezong 德宗 (742–805, reg. 779–805), Emperor Shunzong 順宗 (761–806, reg. 805–806), Emperor Xianzong 憲宗 (778–820, reg. 806–820), Emperor Jingzong 敬宗 (809–827, reg. 824–827), and Emperor Wenzong 文宗 (809–840, reg. 826–840), attempted to reinforce control over the monastic community. They continued to implement policies, including scripture recitation examinations for ordination qualifications and measures like the purification of monks and nuns. While these policies and measures aimed at improving the quality of the Buddhist sangha and increasing literacy, most of them were abandoned shortly after implementation. Occasionally, policies involving the sale of ordination certificates were even introduced. For instance, during the reign of Emperor Muzong 穆宗 (795–824, reg. 820–824), the military commissioner of Xuzhou 徐州, Wang Zhixing 王智興 (758–836), established an Ordination Platform 戒壇 in Sizhou 泗州, conducting ordination ceremonies for those seeking monastic life. This led to a phenomenon where families with three sons would inevitably have one of them ordained 戶有三丁,必令一丁落發[44]. Consequently, during this period, the size of the monastic community continued to expand. By the time Emperor Wuzong 武宗 (814–846, reg. 840–846) initiated the Buddhist suppression, the total number of the sangha reached an astonishing 260,000, constituting a significant portion of the national population, rising from 1 in 400 during the Xuanzong period to 1 in 100[45]. Among them, a substantial number were likely illiterate monks. Due to the lack of precise data, an estimation of the literacy rate during this period remains challenging. However, it is evident that the literacy rate within the monastic community at this time was likely lower than during the Suzong era.

In the Huichang Buddhist Persecution 會昌法難, Emperor Wuzong allowed only a very limited number of temples and monks to remain in Chang'an and certain regions. These monks were mostly elderly and experienced, likely to be literate. After Emperor Wuzong's death, his immediate successor, Emperor Xuanzong 宣宗 (810–859, reg. 846–859), despite adopting an administrative style opposing Wuzong[46], initially implemented relatively strict measures to restrict Buddhism. Decrees issued during Emperor Xuanzong's reign indicate that, until the sixth year of Dazhong 大中 (852), fewer than a thousand old temples were permitted to be reestablished nationwide, with approximately 30,000 monks and nuns ordained. The majority of these monks had been forcibly returned to lay life dur-

ing the Huichang Persecution, and they played a dominant role in the revival of Buddhism from the Dazhong period to the Xiantong 咸通 period. Stringent restrictions on Buddhism persisted until the twelfth month of 852, marking eight years since the Huichang suppression. During this period of strict state control, the overall quality of the Buddhist sangha improved significantly, notably reflected in a substantial increase in literacy rates.

During the reigns of subsequent emperors such as Emperor Yizong 懿宗 (833–873, reg. 859–873) and others following Emperor Xuanzong, the Buddhist monastic community experienced a rapid expansion, accompanied by a gradual rise in the number of inadequately qualified monks, leading to an apparent decline in the literacy rate of the monastic community. In the Central Plains region, the lack of direct materials prevents a direct estimation of the literacy rate. However, considering that, since the year 848, when Dunhuang returned to Tang control under the authority of the Zhang family's Guiyi Army 張氏歸義軍 (848–914), and Buddhist development in Dunhuang began to synchronize with that in the Central Plains, the literacy rate of the monastic community in Dunhuang might offer some insight into the literacy rate of the Central Plains monastic community.

During the rule of the Guiyi Army, the monastic community in Dunhuang also experienced rapid expansion, growing from 427 individuals in the early ninth century (S. 5676) to 1600 individuals in the early tenth century (P. 2704). Among them, there were undoubtedly numerous inadequately qualified monks. This development in the Dunhuang Buddhist monastic community appears to be connected to the Buddhist policies in the Central Plains. According to research, despite the monastic community's malignant expansion, the existence of internal educational mechanisms within the monastic community ensured that a certain proportion of monks and nuns progressed from illiteracy to literacy, estimating this ratio to be approximately between 50% and 65%[47].

The above estimation of the literacy rate of the Tang Dynasty Buddhist monastic community is based on extant historical records, supplemented by certain Dunhuang and Turfan documents. Relevant fluctuations can be graphically illustrated in a chart (see Figure 1):

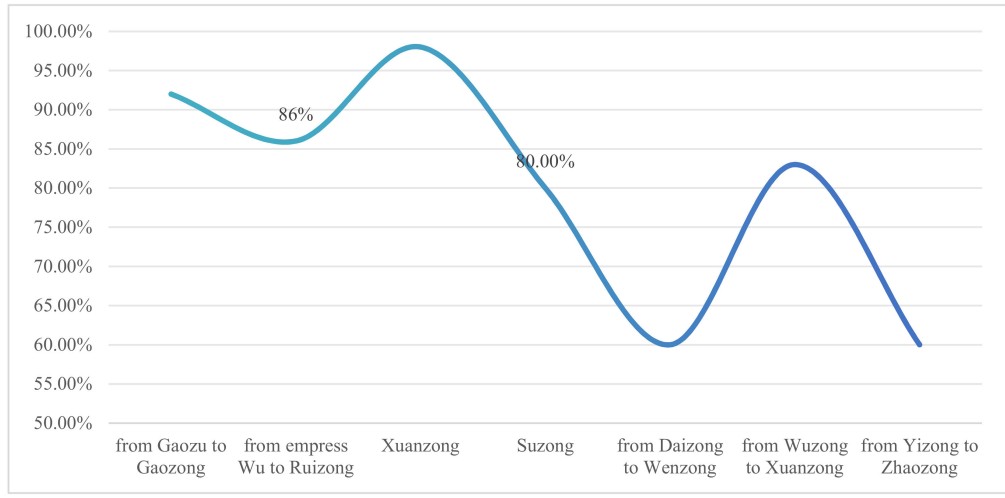

**Figure 1.** Fluctuations in the Literacy Rate of Buddhist Sangha in the Tang Dyansty.

## 4. Conclusions

The Buddhist sangha constituted a significant cultural community in ancient China, exerting substantial influence on the society of the time. The extent of their impact was largely derived from their cultural knowledge. The high or low cultural quality of the sangha not only directly influenced the development of Buddhism itself but also had a profound impact on the broader society in which they existed.

In the history of Chinese Buddhism, particularly in the development of doctrinal studies, during the early Tang Dynasty, eight major Buddhist sects experienced simultaneous prosperity, marking an undeniable peak in the development of Chinese Buddhism. Throughout this developmental phase, eminent monks like Xuanzang 玄奘 (602–664) and

Yijing 義淨 (635–713) played crucial guiding roles. The acceptance and assimilation of Buddhist doctrines, especially those of the Yogācāra school 唯識宗, were evidently closely linked to the overall cultural quality of the monastic community at that time. The high literacy rates within the sangha and the state-sponsored systems such as scripture recitation examinations undoubtedly served as significant safeguards, continually producing eminent monks and providing favorable conditions for the advancement of Buddhist doctrinal studies.

In the middle to later period of the Tang Dynasty, the vibrant scenes of monks traveling nationwide for study and hundreds or thousands of monks attending lectures given by a single master became increasingly rare. During this time, there was limited enthusiasm and involvement in profound Buddhist doctrines, including those associated with the Yogācāra school. The development of Buddhist doctrinal studies entered a relatively subdued phase, not solely due to factors such as the decline of Indian Buddhism or the turmoil in China, but likely influenced by the reduction in the literacy rate within the monastic community. In the later Tang Dynasty, the declining literacy rate objectively hindered monks from delving into profound philosophical doctrines like Yogācāra. Instead, they focused on learning knowledge and ritual skills associated with sects that had lower cultural literacy requirements, known as "expedient means" (方便法門). This phenomenon contributed to the growth of sects such as Pure Land Buddhism 淨土宗. In the context of Central Plains Buddhism, after the later Tang Dynasty, both Pure Land and Chan Buddhism 禪宗 thrived, with practitioners emphasizing the chanting of Buddha names and engaging in skillful question-and-answer sessions. This period did not witness the simultaneous prosperity of the eight major sects, as seen in the early Tang Dynasty.

The fluctuation in the monastic community's literacy rate, particularly its decline, while not significantly impacting the Tang Dynasty's political governance due to the presence of elite monks in the court, still exerted influence on the interplay between politics and religion. Illiteracy often provided lay rulers with a pretext to oppose or regulate Buddhism. Emperor Wu of Northern Zhou, for instance, sought to purge the monastic community, citing monks' insufficient reading abilities. During Emperor Dezong's reign in the Tang Dynasty, Pang Yan 彭偃 (?–784) also appealed for imperial intervention to rectify the monastic community, citing the prevalence of uncultured monks[48]. The cultural quality of the Buddhist monastic community had a more pronounced impact on local governments. Taking Dunhuang as an example, Prof. Rong Xinjiang's research indicates that since the Tubo rule, the growing influence of the Buddhist sangha empowered top-ranking monks to co-govern Dunhuang alongside local authorities. It was not until Zhang Chengfeng's 張承奉 (?–914) rule that the Guiyi Army's political power surpassed religious authority. Prof. Rong contends that this phenomenon is significantly linked to the cessation of Buddhist doctrinal studies in Dunhuang, the decline in the cultural quality of monks and nuns, and the proliferation of inadequately qualified monastics within the sangha[49].

The decline in the literacy rate within the monastic community evidently carries significant adverse implications. However, when observed against the backdrop of monks increasingly residing in lay households rather than monasteries during the later stages of the Tang Dynasty and the growing trend of layization in Buddhism[50], the significance of literate monks within regional societies can, to some extent, also be construed as having positive implications.

The Dunhuang document P. 3608V contains a memorial that mentions Xianyu Shuming 鮮于叔明 (693–787), Linghu Huan 令狐峘 (?–805) and others requesting the emperor to conduct scripture recitation examinations for monks and nuns and to prohibit them from engaging in commercial activities. According to this memorial, during the reign of Emperor Dezong, many temples were occupied by the military, and all temples lacked dining facilities, providing no food for the monks 所在伽藍，例無飯僧. In temples without dining facilities, monks, naturally, could not reside there for an extended period. This is consistent with the characteristics of Dunhuang monasteries where they only "provide food when they have an event" 有事供粮[51]. Before the Huichang Persecution, the Japanese monk

Ennin 圓仁 (793–864) also observed the phenomenon of "monks all living in lay homes" 僧盡在俗家 in the Beihai County of Shandong and other places[52]. This indicates that the phenomenon of monks residing in lay households rather than temples was already quite prevalent before the Huichang Persecution.

From the reign of Emperor Wuzong to the early period of Emperor Xuanzong's rule, approximately 200,000 monks and nuns were forcibly returned to lay life and resided in households[53]. Even if these monks and nuns later regained qualification through ordination, they likely maintained closer connections with lay families. Some of these individuals might have received cultural education in the monasteries before returning to lay life. Based on the estimation in this article, even with a conservative literacy rate of 50% during this period, it implies that there were around 100,000 literate monks and nuns among them. If we adopt the scholarly estimate of an approximately 10% literacy rate in ancient times[54], then during the Huichang and Dazhong eras, an estimated two million literate individuals existed nationwide[55]. The literate monks and nuns residing in lay households constituted one-twentieth of the total literate population, signifying a substantial proportion. While documentation on the interactions between monks and nuns in lay households and the surrounding laypeople is scarce, these monastics significantly assimilated into lay society, a scenario that is not challenging to envision. Therefore, if the Huichang suppression by Wuzong and the restrictions by Xuanzong on Buddhism had a profound impact on the development of Buddhist doctrinal studies, the suppression and restrictions, in turn, impelled literate monks and nuns into lay society. This process facilitated the dissemination of knowledge from within monasteries to the broader society, evidently contributing positively to local social development[56].

**Funding:** This research was funded by Key grant Project of Chinese Ministry of Education grant number 22JZD027 and Shandong Provincial Social Science Foundation project grant number 22CLSJ05.

**Data Availability Statement:** All the data are calculated in this article, and there is no link.

**Conflicts of Interest:** The author declares no conflict of interest.

## Notes

1. In the several centuries following the establishment of Buddhism by Siddhartha Gautama in the 6th century BCE, the literacy of monks was not a pressing concern. During this period, Buddhist scriptures primarily relied on oral transmission through memorization. It was not until around the turn of the Common Era that the practice of recording Buddhist scriptures in written form gradually emerged. However, even with this development, whether in India or in regions such as Central Asia and Southeast Asia, there was not a strong imperative for monks to be literate. This stands in marked contrast to the development of Buddhism in China.

2. In *The Ten Precepts and Conduct Rules Observed by Novices* (沙彌十戒法并威儀), it is stipulated that novices are required to recite scriptures three times daily 晝夜三時,誦經行道. see T 1471: 932a,b.

3. T 396: 1119a.

4. T 2012: 85a.

5. T 2103: 279a,b.

6. T 2060: 565c.

7. Before the 8th year of the Tianbao era (749), the Tang government stipulated the "registration of the sangha every three years" 三年一造. Subsequently, after this period, it was revised to "register the sangha every ten years" 諸州僧尼籍帳等，每十年一造. Following the An Lushan Rebellion, although the compilation of sangha registrations became somewhat disordered, the state continued to implement periodic reorganization measures. For instance, in the 4th year of the Taihe era during the reign of Emperor Wenzong (830), it was mandated that "the sangha registrations of local prefectures and jurisdictions…be compiled every five years" 諸州府僧尼籍帳，今五年一造. See: *Tang liudian* 4: 126. *Cefu yuangui* 474: 5369.

8. Regarding the research on the ordination system, see (Michihata 1967, pp. 29–177; Zhan 1998, pp. 5–13; Ming 2003, pp. 179–92; Lai 2010, pp. 120–206).

9. Regarding the research on the *shijing* in Tang dynasty, see (Bai 2005, pp. 31–36).

10. The Tang emperors' choice to adopt the scripture recitation examination as the primary method for ordaining monks and nuns might be a continuation of the policies established during the periods of Emperor Wu of Northern Zhou and the Last Emperor of the Chen Dynasty. It is also plausible that the influence of the imperial examination system played a role.

11. T 2053: 275c.

12     T 2037: 822c.

13     T 2035: 452c.

14     T 2035: 379a.

15     *Tang huiyao*: 861.

16     *Song gaoseng zhuan*: 374.

17     *Tang da zhaoling ji*: 591.

18     *Song gaoseng zhuan*: 736.

19     *Cefu yuangui* 474: 5369.

20     *Wudai huiyao*: 153–154.

21     *Tang huiyao*: 860.

22     See (Dai 2006, pp. 105–32).

23     About the register, see (Meng 2007, pp. 50–55; Meng 2009, pp. 136–43).

24     *Tang huiyao*: 861.

25     About the *Qingdi shangzuo*, see (Wu 2022a, pp. 25–47).

26     *Zizhi tongjian* 245: 7906.

27     Here, I would like to express my gratitude to an anonymous reviewer for providing insightful suggestions. The reviewer expressed concerns about the use of the scripture recitation examination as a criterion for judging literacy, stating that "a monk may conceivably have been 'literate' enough to follow the text of a scripture while chanting it, but not be able to write a simple legal contract". Indeed, the extant materials from the 7th to 8th centuries preserved in classical literature are not rich enough to fully address this issue. However, the Dunhuang materials preserved from the 8th to 10th centuries can demonstrate that literate monks during that time were capable of reading scriptures to fulfill religious duties while also using their cultural knowledge to engage in activities requiring written skills to assist the lay community. Additionally, the question raised by the reviewer touches upon a topic that scholars often debate when studying literacy rates, namely the issue of full literacy and functional literacy. In my view, the requirement of the scripture recitation examination entails monks and nuns reading tens of thousands of characters from Buddhist scriptures, which suggests that monks and nuns who have passed the examination are likely acquainted with a considerable number of written characters, classifying them as possessing full literacy rather than functional literacy. For discussions on this topic, see (Solomon 1971; Mote 1972; Idema 1974; Rawski 1979; Idema 1980).

28     *Tulufan chutu wenshu* 8: 485–491.

29     *Jiu Tangshu* 89: 2893.

30     See (Zhou 2008, p. 15).

31     *Cefu yuangui* 159: 1775.

32     *Jiu Tangshu* 96: 3022.

33     *Jiu Tangshu* 8: 172.

34     *Tang huiyao*: 837.

35     *Quan Tangwen buyi*: 104–105.

36     *Fayuan zhulin*: 2898.

37     *XinTangshu* 125: 4398.

38     *Jiu Tangshu* 89: 2893.

39     Regarding the content of this stele inscription, refer to Quan Tangwen 987: 10209. for further study on this stele inscription, consult (Wu 2022a, pp. 25–47).

40     *Tongdian* 11: 244.

41     *Xin Tangshu* 51: 1347.

42     *Tang liudian*: 125.

43     See (Wu 2022b, pp. 1–25).

44     *Jiu Tangshu* 174: 4514.

45     See (Xie 2009, p. 437).

46     *Zizhi tongjian* 248: 8029–8030.

47     See note 43 above.

48     *Jiu Tangshu* 127: 3580.

49     See (Rong 2015, p. 275).

50     For the reasons behind monks residing in lay households, refer to (Wu 2018, pp. 14–21).

51     See (Hao 1998, pp. 123–63).

52    *Nittō-guhō-junrei-kōki no kenkyū*: 228–349.

53    According to research, from the comprehensive Buddhist suppression initiated by Emperor Wuzong in April of 845 until the complete lifting of restrictions on Buddhism by Emperor Xuanzong in the twelfth month of 852, a period of eight years, less than a thousand old temples were permitted to be reconstructed nationwide, and approximately 30,000 monks and nuns were ordained. Refer to (Wu, forthcoming).

54    See (Goody 1963, pp. 304–45; Baines 1983, pp. 572–99; Harris 1989; Beard et al. 1991).

55    During the reign of Emperor Wuzong, the total number of households in the realm was 4,955,151, with an estimated population of around 20 million. Regarding the household data, see *Cefu Yuangui* 486: 5515.

56    In Buddhist monasteries during the medieval period, monks not only imparted Buddhist knowledge but also provided education in secular subjects. Numerous documents, written by monks and including items such as divorce letters, property division documents, testaments, dividing family inheritance documents, are preserved in the Dunhuang caves. These documents cover virtually every aspect of daily life for ordinary people that requires written records. They convincingly demonstrate that during the medieval period, monks were actively involved in various aspects of lay society. For further insights into the education of secular knowledge within monasteries, see (Mair 1981; Galambos 2015).

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
