# Peer review of "A Study on the Literacy Rate of Buddhist Sangha in the Tang Dynasty"

_religions, doi:10.3390/rel15030306_

Round 1

Reviewer 1 Report

Comments and Suggestions for Authors

The author has an excellent command of the main sources of information, and these have been presented conscientiously.  But there appears to be no discussion of a fundamental issue, namely the exact nature of ‘literacy’ within the Buddhist context.  The Buddha himself was after all most likely illiterate, and in South Asia Buddhist materials were for the most part transmitted orally until about two thousand years ago; the strength of this oral tradition of memorization remained important and it was certainly the source of some translations into Chinese.  In ‘Mahayana’, and certainly in China, the materiality of the text became important and fostered a literate scribal culture, but it is unclear what role memorization in the absence of text continued to play.  This may complicate what the sources appear to tell us, because it seems it was possible to ‘know’ and expound a text without being able to write it out: how about the story of 慧能, for example? 

Comments on the Quality of English Language

The English is grammatically fine, but some conventions of transcription and citation are frequently ignored.

1.        All names of monks, at least after about 400 CE, are personal names, so the transcriptions should be concatenated: Tanji , Zhiyi 智𫖮.

2.       era’ in English is a standard translation of 年号, so while it is fine in that meaning, it cannot be used as a synonym for ‘reign’   during the Tang period.

3.       Non-Chinese names are mutilated: ‘Sando’ is the Japanese for sandwich, not Michihata Ryōshū 道端良秀; ‘Jack’, ‘John’, ‘William’, ’Mary’, etc. should be ‘Goody’, ‘Baines’, ‘Beard’, ‘Harris’.

4.       Nobody outside China knows what ‘Tubo’ means as a Chinese translation of bod chen po , so this needs to be explained.

Author Response

Thank you very much for the thorough and professional feedback. I have meticulously revised the paper based on your suggestions. Below is some feedback addressing the specific issues you raised.

Q 1: But there appears to be no discussion of a fundamental issue, namely the exact nature of ‘literacy’ within the Buddhist context.  The Buddha himself was after all most likely illiterate, and in South Asia Buddhist materials were for the most part transmitted orally until about two thousand years ago; the strength of this oral tradition of memorization remained important and it was certainly the source of some translations into Chinese.  In ‘Mahayana’, and certainly in China, the materiality of the text became important and fostered a literate scribal culture, but it is unclear what role memorization in the absence of text continued to play.  This may complicate what the sources appear to tell us, because it seems it was possible to ‘know’ and expound a text without being able to write it out: how about the story of 慧能, for example?

R 1: This is a thought-provoking question. The requirement for monks to be literate is a highly specific phenomenon. In the several centuries following the establishment of Buddhism by Siddhartha Gautama in the 6th century BCE, the literacy of monks was not an urgent concern, as Buddhist scriptures primarily relied on memorization. It wasn’t until around the turn of the Common Era that the trend of recording Buddhist scriptures in written form gradually emerged. However, even with this development, whether in India or in regions such as Central Asia and Southeast Asia, there was not a strong imperative for monks to be literate. This contrasts significantly with the development of Buddhism in China. I have added a footnote in the article to provide a brief response to this question raised by the reviewer.

Q 2: All names of monks, at least after about 400 CE, are personal names, so the transcriptions should be concatenated: Tanji 曇积, Zhiyi 智顗.

R 2: Yes, I have made revisions in accordance with the reviewer’s suggestions.

Q 3: ‘era’ in English is a standard translation of 年号, so while it is fine in that meaning, it cannot be used as a synonym for ‘reign’   during the Tang period.

R 3: Yes, I have made revisions in accordance with the reviewer’s suggestions.

Q 4: Non-Chinese names are mutilated: ‘Sando’ is the Japanese for sandwich, not Michihata Ryōshū 道端良秀; ‘Jack’, ‘John’, ‘William’, ’Mary’, etc. should be ‘Goody’, ‘Baines’, ‘Beard’, ‘Harris’.

R 4: The reviewer’s suggestions are valid, and I have made the necessary revisions. I apologize for such a basic mistake.

Q 5: Nobody outside China knows what ‘Tubo’ means as a Chinese translation of bod chen po , so this needs to be explained.

R 4: Yes, I overlooked that. Now, I provided an explanation the first time I used the term 'Tubo' in the text: Tubo was an empire established by Tibetans from the early 7th century to the mid-9th century, and it ruled Dunhuang from 786 to 848.

Reviewer 2 Report

Comments and Suggestions for Authors

there are three major issues in this paper. this paper can't be published without resolving these issues.

first of all, it is not sufficient in consulting secondary sources. there are some other Western-language sources that this paper should consult, such as Erik Zurcher's paper on the Buddhist education in the Tang dynasty, Imre Galabos's papers on manuscripts and writing style, and Christoper Nugent's book, et al.

the second big issue is the translation problem. numerous technical terms in this paper are often wrongly translated, or very unprofessional, which should not appear in an academic paper. for example, it translates su俗 as secular, which is wrong. it should be translated as "lay." the jietan 戒壇 should be translated into the ordination platform (see Funayama Toru, Huaiyu Chen, and John McRae's translations). the Chinese character zhi 紙 should be translated as sheet, especially it was mentioned as the number of sheets for copying Buddhist scriptures (see Imre Galambos, Jean-Pierre Drege's papers and books). fangbian famen 方便法門was translated into simplfied methods, which is wrong, not a Buddhist translation. I also found that many translations of primary sources from both traditional literature and Dunhuang manuscripts are not accurate. they often missed one or two sentences, not closely following the original passages. Please double check these cited passages and re-translate them. the current translations are not acceptable. I would suggest that all translations and technical terms should be re-examined.

the third issue is the style. the paper is not consistent in style. for example, many names did not include dates. and some names appeared different from standard style, such as Zhi Yi, which should appear as Zhiyi. Please double check the style. 

Comments on the Quality of English Language

re-working on translations and technical terms

Author Response

Thank you very much for the thorough and professional feedback. I have meticulously revised the paper based on your suggestions. Below is some feedback addressing the specific issues you raised.

Q 1: it is not sufficient in consulting secondary sources. there are some other Western-language sources that this paper should consult, such as Erik Zurcher's paper on the Buddhist education in the Tang dynasty, Imre Galabos's papers on manuscripts and writing style, and Christoper Nugent's book, et al.

A 1: The reviewer’s suggestions are highly professional. Erik Zurcher was one of the most prominent scholars in the study of Chinese Buddhism, and the achievements of Imre Galabos and Christopher Nugent in Dunhuang research are also noteworthy. I have read many of their works and articles, but unfortunately, I neglected to reference them while writing this paper. I have translated Erik Zurcher’s “Buddhism in China Collected Papers of Erik Zurcher” into Chinese, and it will be published soon. Therefore, I am quite familiar with his research on Buddhist education in the Tang Dynasty. He utilized numerous historical sources and Dunhuang documents to study Buddhist education, producing a very significant outcome. Moreover, I have cited this particular work in a previously published article on the literacy rate of the Dunhuang Sangha. Consequently, the omission of this reference in the bibliography is indeed inappropriate.

Q 2: the second big issue is the translation problem. numerous technical terms in this paper are often wrongly translated, or very unprofessional, which should not appear in an academic paper. for example, it translates su俗 as secular, which is wrong. it should be translated as "lay." the jietan 戒壇 should be translated into the ordination platform (see Funayama Toru, Huaiyu Chen, and John McRae’s translations).

A 2: Yes, I have made revisions in accordance with the reviewer’s suggestions.

Q 3: the Chinese character zhi 紙 should be translated as sheet, especially it was mentioned as the number of sheets for copying Buddhist scriptures (see Imre Galambos, Jean-Pierre Drege's papers and books).

A 3: I have read articles by Imre Galabos and Christopher Nugent on Dunhuang manuscripts and literary works, but I did not pay attention to the terminology used in them, leading to some elementary mistakes. I sincerely apologize. Following the recommendations of the reviewer, I have revised some of these errors.

Q 4: fangbian famen 方便法門was translated into simplfied methods, which is wrong, not a Buddhist translation.

A 4: I replaced “Expedient Means” with “simplified methods.”

Q 5: I also found that many translations of primary sources from both traditional literature and Dunhuang manuscripts are not accurate. they often missed one or two sentences, not closely following the original passages. Please double check these cited passages and re-translate them. the current translations are not acceptable. I would suggest that all translations and technical terms should be re-examined.

A 4: Indeed, I have made some abridgments in certain quotations. However, from my perspective, these deletions were deemed necessary to present the desired content in a clear and concentrated manner without altering the original meaning of the passages. The intention behind these modifications was to enhance the clarity and focus of the presented material while preserving the integrity of the original paragraphs. I am open to further discussion and can provide additional justifications for any specific deletions in order to address concerns about accuracy and completeness.

Q 6: the third issue is the style. the paper is not consistent in style. for example, many names did not include dates. and some names appeared different from standard style, such as Zhi Yi, which should appear as Zhiyi. Please double check the style.

A 6: I appreciate your observation regarding the inconsistency in the spelling of names. I have duly rectified this issue to ensure uniformity across the paper. Regarding the temporal information of the individuals mentioned in this paper, I have now included specific historical periods wherever possible, thereby offering a clearer context for their lives.

Reviewer 3 Report

Comments and Suggestions for Authors

I think this paper can be published, but please address the following.

This paper needs to be formatted properly. The Chinese citations are not in block quotes, so it looks like the quotes of 冊府元龜 and other texts are the author's own words.

"Erik Zürcher once described the an-cient Buddhist sangha as “the Secondary Elite.”(Erik 1989)."

Zürcher is the surname: Zürcher 1989

"In the several centuries following the establishment of Buddhism by Siddhartha Gautama in the 6th century BCE,"

The date of 6th century BCE is something of a guess.

"the culling of monks"

Culling implies killing. The right verb is "defrocking" I think.

"even boasted that he had “not recognized characters since birth.” 一生以來,不識文字"

It wasn't boasting. Huineng in the story simply says he can't read. He wasn't proud or ashamed of it.

Also, this was not a historical account. The 六祖壇經 is a fictional account, not a historical account. Huineng also was never trained as a monk. He simply went to the temple and was working there without any formal schooling.

The Vinaya doesn't require reading strictly speaking. In fact, the Vinaya ordination assumes the prospective novice knows not much at all, which is why they are trained.

Tubo rule = 吐蕃 is normally rendered Tibet in English language scholarship. If you want to use Tubo instead of Tibet, you should add a footnote.

"This suggests that even if all these individuals were illiterate, the lit-eracy rates at that time might have been approximately 80.8% for monks and 83.6% for nuns."

I am not entirely clear on how the author gets to this percentage, but if the author wants to stand by this theory, then I am fine with it so long as the formula is shown.

"the literacy rate of the monastic community in Dunhuang might offer some insight into the literacy rate of the Central Plains monastic community."

One could argue otherwise: that Dunhaung was remote and not necessarily so well educated as Chang'an. That would be like comparing the literacy rates at Hadrian's Wall in the Roman empire with those of Rome.

"张承奉" = use 繁体字 (张 = 張)

The following study might be of interest, since it shows that very few monks in East Asia could read Sanskrit:

Kotyk, Jeffrey. 2021. “The Study of Sanskrit in Medieval East Asia: China and Japan.” Hualin International Journal of Buddhist Studies 4, no. 2: 240–273. 

Comments on the Quality of English Language

Please fix all the block quotes! Like this:

Eevery three years, a comprehensive census of all Taoists, female Taoists, monks, and nuns was to be conducted at the state and county levels, resulting in the compilation of a register. The register was required to include information such as the time of their monas-tic qualification, duration of monastic life, and academic achievements. The seals of the state and county were to be affixed to the designated positions on the register.22

諸道士、女冠、僧、尼,州縣三年一造籍,具言出家年月、夏臘、學業,隨處印署。

Make it into a block quote:

Every three years, a comprehensive census of all Taoists, female Taoists, monks, and nuns was to be conducted at the state and county levels, resulting in the compilation of a register. The register was required to include information such as the time of their monastic qualification, duration of monastic life, and academic achievements. The seals of the state and county were to be affixed to the designated positions on the register. 諸道士、女冠、僧、尼,州縣三年一造籍,具言出家年月、夏臘、學業,隨處印署。

Author Response

Thank you very much for the thorough and professional feedback. I have meticulously revised the paper based on your suggestions. Below is some feedback addressing the specific issues you raised.

Q 1: "Erik Zürcher once described the an-cient Buddhist sangha as “the Secondary Elite.”(Erik 1989)." Zürcher is the surname: Zürcher 1989

A 1: Yes, I have made revisions in accordance with the reviewer’s suggestions.

Q 2: "In the several centuries following the establishment of Buddhism by Siddhartha Gautama in the 6th century BCE,"

The date of 6th century BCE is something of a guess.

A 2: Yes, there is still debate over the exact dates of Siddhartha Gautama's life, but the notion of the 6th century BCE seems to have gained a fair amount of acceptance. I haven’t delved deeply into this issue myself, so I’ve opted for a fairly conventional viewpoint here.

Q 3: "the culling of monks"

Culling implies killing. The right verb is "defrocking" I think.

A 3: Yes, the verb suggested by the reviewer is more accurate. I have revised it accordingly.

Q 4: "even boasted that he had “not recognized characters since birth.” 一生以來,不識文字"

It wasn't boasting. Huineng in the story simply says he can't read. He wasn't proud or ashamed of it. Also, this was not a historical account. The 六祖壇經 is a fictional account, not a historical account. Huineng also was never trained as a monk. He simply went to the temple and was working there without any formal schooling.

A 4: Yes, indeed, the term “boast” I used does not quite align with the context of the 六祖壇經. I have replaced it with a more neutral term, “said.” While the depiction of Huineng as illiterate in the Platform Sutra may not be historically accurate. I intend to use Huineng’s perspective to observe a prevailing attitude of the time, namely, whether illiteracy was perceived as an obstacle to becoming a monk.

Q 5: The Vinaya doesn't require reading strictly speaking. In fact, the Vinaya ordination assumes the prospective novice knows not much at all, which is why they are trained.

A 5: Yes, according to the Vinaya, this understanding can indeed be derived. The Vinaya does require monks to be literate.

Q 6:  Tubo rule = 吐蕃 is normally rendered Tibet in English language scholarship. If you want to use Tubo instead of Tibet, you should add a footnote.

A 6: Yes, Yes, that was my oversight. I have added a sentence to clarify in the text: “Tubo was an empire established by Tibetans from the early 7th century to the mid-9th century, and it ruled Dunhuang from 786 to 848.”

Q 7: "This suggests that even if all these individuals were illiterate, the lit-eracy rates at that time might have been approximately 80.8% for monks and 83.6% for nuns."

I am not entirely clear on how the author gets to this percentage, but if the author wants to stand by this theory, then I am fine with it so long as the formula is shown.

A 7: The data was obtained through statistical analysis of P. 4072(3), and the specific argument is as follows:

 Dunhuang Document P. 4072(3), titled “Report on the Sale of Tonsure Certificates in Hexi by Zhang Jiali in the Second Year of Qianyuan (759),” 唐乾元二年(759)張嘉禮河西六州納錢度僧告牒 records a total of 327 monks and 169 nuns receiving monastic ordination in the six provinces of Hexi. By the end of the Kaiyuan era, there were 3,245 monasteries nationwide with 75,524 monks and 2,113 nunneries with 50,576 nuns. With 328 provinces in the country, this averages to 10 monasteries per province with an average of 23 monks and 6 nunneries per province with an average of 24 nuns. Therefore, in the six provinces of Hexi, there were approximately 60 monasteries, 36 nunneries, 1,380 monks, and 864 nuns. Thus, the number of monks and nuns who obtained monastic ordination through monetary donations in 759 accounted for around 19.2% (327/1,707) and 16.4% (169/1,033) of the total population in monasteries and nunneries, respectively.

Since the proportion of illiterate monks and nuns may be 19.2% and 16.4% respectively, the proportion of literate monks and nuns would then be 80.8% and 83.6% respectively.

Q 8: "the literacy rate of the monastic community in Dunhuang might offer some insight into the literacy rate of the Central Plains monastic community."

One could argue otherwise: that Dunhaung was remote and not necessarily so well educated as Chang'an. That would be like comparing the literacy rates at Hadrian's Wall in the Roman empire with those of Rome.

A 8: This is a fascinating and valuable question, but I have to admit that I still have no direct evidence to answer it, and I hope to further study this issue in depth in the future.

Q 9: Please fix all the block quotes!

A 9: Yes, I have adjusted the format.

Q 10: The following study might be of interest, since it shows that very few monks in East Asia could read Sanskrit:

Kotyk, Jeffrey. 2021. “The Study of Sanskrit in Medieval East Asia: China and Japan.” Hualin International Journal of Buddhist Studies 4, no. 2: 240–273.

A 10: Thank you for recommending the article.

Round 2

Reviewer 1 Report

Comments and Suggestions for Authors

Author Response

Q1: The queries are generally well answered, butwo problems remain here.  The first footnote concedes that adherence to Buddhism does not intrinsically require a literate clergy and that literacy as a requirement is influenced by the Chinese situation. But does the Chinese situation require that all clergy be literate, and if so, why?  Some clergy certainly had to be literate to promote Buddhism with a literate elite.  Daoist clergy also had to be literate, in that they communicated with higher Chinese   divinities that used writing, and this model may have brought pressure to bear on their Buddhist rivals.  Obviously, the piece cannot be rewritten to include any new perspectives on this, but we need a little more than the first footnote.  Otherwise, all learn is that they had to be literate because they    were Chinese, which is a bit mysterious.

A1: It must be acknowledged that I am uncertain about why Emperor Wu of Northern Zhou and the Last Emperor of Chen Dynasty used literacy as a criterion for managing monks and nuns. The Tang Dynasty formally adopted the scripture recitation examination as the primary means of ordaining monks and nuns, and the reasons for this decision might be a continuation of the policies established during the periods of Emperor Wu of Northern Zhou and the Last Emperor of Chen Dynasty. It is also plausible that the influence of the imperial examination system played a role. As for the potential impact of Taoist factors, I have not conducted an in-depth study. However, the reviewer’s perspective is enlightening, and I may explore this line of analysis in future research. For now, in the footnotes, I can briefly introduce possible reasons for the Tang emperors’ decision:

The Tang emperors’ choice to adopt the scripture recitation examination as the primary method for ordaining monks and nuns might be a continuation of the policies established during the periods of Emperor Wu of Northern Zhou and the Last Emperor of Chen Dynasty. It is also plausible that the influence of the imperial examination system played a role.

letters for dividing family inheritance, housewarming and house securing rituals, mourning letters,

Q2: But once this situation is addressed there is also a problem throughout with the unqualified use of a literate/illiterate distinction as though it is a clearcut choice of two options.  A fishmonger may know one hundred written characters for fish but not be able to read a newspaper.  Similarly, a monk may  conceivably have been ‘ literate’ enough to follow the text of a scripture while chanting it, but not be able to write a simple legal contract.  The Dunhuang materials suggest that there were persons in

monasteries who could do the latter, but could every monk who could ‘ read’ the Diamond Sutra, aided by memorization, do that?

One inventory that seeks to identify those who were aspiring to literacy in Dunhuang is Victory Mair, “Lay Students and the making of Written Vernacular Narratives”, Chinoperl Papers 10 (1981),pp. 5-    96, though this only shows that the evidence suggests they were not monks.  Equally copyists 經生   seem to have been lay people.  But some monks at Dunhuang do seem to have used a wide range of written materials, not just scriptures or other prayers for chanting.  The approach used in this paper  provides quantified information worth presenting, but it must be conceded that the sources used,

and deployment of a single category of ‘literacy’, may conceal a complex situation.  Again, the matter cannot be fully discussed without complete rewriting, but it does at least have to be briefly stated.

A2: Yes, the question raised by the reviewer is a pertinent one, addressing the distinction between functional literacy and full literacy. This has been a recurrent topic of debate among scholars studying literacy rates in recent decades. Even with regard to “functional literacy,” there remains a contentious issue regarding the specific number of characters one must recognize to be considered functionally literate and how many characters constitute full literacy. Scholars such as Frederick W. Mote (1972), Evelyn Rawski (1979), among others, have engaged in discussions on these issues in the context of modern literacy studies. Despite the wealth of modern historical materials at our disposal compared to ancient sources, achieving consensus among scholars remains challenging. Therefore, it is evident that addressing this issue in the study of ancient literacy rates poses a significantly greater challenge.

In fact, I have previously discussed the above content in a dedicated study on the literacy rates of Dunhuang monks from the 8th to the 10th centuries. However, the omission of this discussion in the current article has indeed resulted in a deficiency in the argumentation. I would like to express gratitude to the reviewer for presenting this highly professional and insightful question. I have supplemented the footnotes with discussions on relevant scholarly debates and my stance on these issues.

Footnote 1

Here, I would like to express my gratitude to an anonymous reviewer for providing insightful suggestions. The reviewer expressed concerns about the use of the scripture recitation examination as a criterion for judging literacy, stating that “a monk may conceivably have been ‘literate’ enough to follow the text of a scripture while chanting it, but not be able to write a simple legal contract.” Indeed, the extant materials from the 7th to 8th centuries preserved in classical literature are not rich enough to fully address this issue. However, the Dunhuang materials preserved from the 8th to 10th centuries can demonstrate that literate monks during that time were capable of reading scriptures to fulfill religious duties while also using their cultural knowledge to engage in activities requiring written skills to assist the lay community. Additionally, the question raised by the reviewer touches upon a topic that scholars often debate when studying literacy rates, namely the issue of full literacy and functional literacy. In my view, the requirement of the scripture recitation examination entails monks and nuns reading tens of thousands of characters from Buddhist scriptures, which suggests that monks and nuns who have passed the examination are likely acquainted with a considerable number of written characters, classifying them as possessing full literacy rather than functional literacy. For discussions on this topic, see Mote( 1972), Evelyn( 1979).

Footnote 2

In Buddhist monasteries during the medieval period, monks not only imparted Buddhist knowledge but also provided education in secular subjects. Numerous documents, written by monks and including items such as divorce letters, property division documents, testaments, dividing family inheritance documents, are preserved in the Dunhuang caves. These documents cover virtually every aspect of daily life for ordinary people that requires written records. They convincingly demonstrate that during the medieval period, monks were actively involved in various aspects of lay society. For further insights into the education of secular knowledge within monasteries, see Victor Mair (1981) and Imre Galambos (2015).

Reviewer 2 Report

Comments and Suggestions for Authors

please consult with the editor

Comments on the Quality of English Language

another check 

Author Response

The reviewer appears not to have uploaded specific recommendations.

In the new version, I have added relevant studies by Western scholars.